# Age is just a number: Examining the preservation of cells and soft tissues in *Bothriolepis* and other Devonian fish

Christopher L. Rogoff[1], Paul V. Ullmann[ID][2]*

**1** Department of Geology, Rowan University, Glassboro, New Jersey, United States of America, **2** Harold Hamm School of Geology and Geological Engineering, University of North Dakota, Grand Forks, North Dakota, United States America

* paul.ullmann@und.edu

## Abstract

Recent microscopy and proteomic studies demonstrate fossil bones can preserve remarkable palimpsests of cellular and soft tissue evolution, yet it remains unclear if there are temporal limits to such preservation. For instance, it remains unknown if fossil cells and soft tissues can be recovered from fossil bones of the earliest vertebrates from the middle Paleozoic. To test this, we demineralized nine bone fragments of *Bothriolepis* and other Late Devonian fish. Removal of phosphates from the fossils released numerous microstructures morphologically consistent with vertebrate osteocytes, pieces of blood vessels, and sheets of fibrous bone matrix which were frequently found to exhibit elemental chemistry potentially indicative of partial organic composition. These discoveries extend this style of exceptional preservation to aspidin and dentine and predate all prior reports of similar cellular/soft-tissue preservation in fossil bones by nearly 100 million years, indicating that the geologic age of a fossil specimen is a poor predictor of whether or not it will retain potentially-endogenous microstructures. Thus, the preservation pathways leading to these forms of soft-tissue fossilization likely began contemporaneously with the evolution of vascularized cellular bone in the early Paleozoic.

## Introduction

Though once thought implausible, it is now clear that fossils can preserve exquisite soft tissues not only as mineralized pseudomorphs, but also through retention of endogenous (original) cells, tissues, and their component biomolecules. Recent studies of Mesozoic fossils, for example, have identified preservation of osteocytes [e.g., 1–3], blood vessels [e.g., 4,5], melanosomes [e.g., 6,7], extracellular fibrous matrix [e.g., 1,8], leukocytes [9,10], and chondrocytes [11,12], several of which have yielded original peptide sequences pertaining to collagen I, myosin, actin, beta tubulin, and other eukaryotic proteins [13–16]. In several cases, independent

**Data availability statement:** All relevant data are within the paper and its Supporting information files.

**Funding:** PVU received financial support for this research from Rowan University. The funder did not play any role in the study design, data collection and analysis, decision to publish, or preparation of the manuscript.

**Competing interests:** The authors have declared that no competing interests exist.

replication [e.g., 14,16] and use of complementary techniques [e.g., 17,18] have further supported the endogeneity of many recovered microstructures with biochemical and/or geochemical data, requiring paleontologists to revise historical models of fossilization via mere mineral replication [i.e., 19]. Attempts to elucidate soft tissue preservation mechanisms have yielded intriguing clues into biomolecular stabilization reactions [e.g., 20], yet numerous questions remain about how the sources of ancient biomolecules – cellular and soft-tissue microstructures – are able to persist in geologically-ancient fossils [21].

One question of critical importance is if geologic age bears negligible influence on this style of exceptional preservation, meaning that cells and soft tissue microstructures could be viably recovered from the earliest vertebrates possessing cellular, vascularized bone (i.e., osteostracans in the middle Paleozoic) [22]. Following this hypothesis, cellular and vascular bone of any geologic age could be expected to yield potentially-endogenous osteocytes, blood vessels, and fibrous matrix.

We tested this hypothesis by performing demineralization assays of 361–378-million-year-old fossil bone specimens, each approximately 100 million years older than any others previously analyzed using this technique (the oldest specimens previously tested are from Permian reptiles [22]). Demineralization assays utilize a weak acid to isolate organic microstructures and oxides lacking divalent cations from biomineralized tissues [21], which can then be analyzed by diverse microstructural, histochemical, immunological, and/or proteomic methods to characterize their morphology, elemental composition, and even molecular composition [e.g., 14,15,23,24].

In this study, ~0.5 cm$^3$ fragments of nine Late Devonian fish bones and three accompanying rock matrix samples were demineralized in 0.5 M ethylenediaminetetraacetic acid (EDTA) pH 8.0 (see Materials and methods section below). Recovery of microstructures morphologically consistent with endogenous vertebrate cells (i.e., osteocytes) and/or soft tissues (i.e., blood vessels) from any of these Devonian fossils would support the conclusion that ancient geologic age does not significantly diminish soft tissue or, by extension, biomolecular preservation potential.

## Materials and methods

### Materials

The examined fossils include a fragment of dermal (acellular) bone from the heterostracan *Psammosteus*, two pectoral fin spines of the acanthodian *Gyracanthus*, a jaw fragment of the tristichopterid *Hyneria*, scales of the megalichthyid *Megalichthys* and porolepiform *Holoptychius*, and anteroventrolateral plates of the antiarch "placoderm" *Bothriolepis* and an indeterminate bothriolepid (Table 1 and Tables A and B in S1 File). Most of these fossils were recovered from sandy "red bed" fluviodeltaic siltstones at the famous Red Hill locality and other nearby sites in the Catskill Formation in Pennsylvania, but the indeterminate bothriolepid (NUFV 1586) and *Psammosteus* specimens were recovered from sandy fluvial overbank siltstones of the Fram and Nordstrand Point formations, respectively, in Canada [25,26].

The two Canadian Nunavut specimens examined in this study were collected in 2000 (NUFV 1587) and 2004 (NUFV 1586) by Nunavut Paleontological Expedition

**Table 1. Summary of cellular and soft tissue recovery from nine Devonian fish bones.**

| Specimen # | Major Clade | Minor Clade | Taxon | Skeletal Element | Formation | Locality | 'Osteocytes' | 'Vessels' | 'Fibrous Matrix' |
|---|---|---|---|---|---|---|---|---|---|
| NUFV 1587 | Pteraspidomorphi | Heterostraci | *Psammosteus* sp. | Dermal bone | Nordstrand Point | Canada, NV2K11 | N/A | Rare | Uncommon |
| ANSP uncat. | "Placodermi" | Antiarchi | *Bothriolepis* sp. | Anteroventro-lateral plate | Catskill | Powys Curve | Abundant | Rare | Uncommon |
| NUFV 1586 | "Placodermi" | Antiarchi | Bothriolepidae indet. | Anteroventro-lateral plate | Fram | Canada, NV0403 | Abundant | Rare | Rare |
| ANSP uncat. | Eugnathostomata | "Acanthodii" | *Gyracanthus sherwoodi* | Pectoral fin spine | Catskill | Red Hill | Abundant | Frequent | Rare |
| ANSP uncat. | Eugnathostomata | "Acanthodii" | *Gyracanthus sherwoodi* | Pectoral fin spine | Catskill | Red Hill | Abundant | Frequent | Absent |
| ANSP 23543 | Osteichthyes | Porolepiformes | *Holoptychius* sp. | Scale | Catskill | Tioga Rest Stop | Uncommon | Abundant | Rare |
| ANSP 21165 | Osteichthyes | Megalichthyidae | *Megalichthys mullisoni* | Scale | Catskill | Red Hill | Abundant | Uncommon | Rare |
| ANSP 21165 | Osteichthyes | Megalichthyidae | *Megalichthys mullisoni* | Scale | Catskill | Red Hill | Abundant | Uncommon | Rare |
| ANSP 25034 | Osteichthyes | Tristichopteridae | *Hyneria lindae* | Scale | Catskill | Red Hill | Abundant | Uncommon | Rare |

Specimens are listed in approximate evolutionary order according to the first appearance of the major clade to which they belong. Abbreviations: ANSP, Academy of Natural Sciences of Drexel University (formerly "of Philadelphia"); NUFV, Nunavut Fossil Vertebrate Collection at the Canadian Museum of Nature in Ottawa, Ontario, Canada; uncat, uncatalogued.

crews, and the seven specimens from Red Hill and other nearby sites in Pennsylvania were collected between 1994 and 2018 by Academy of Natural Sciences of Philadelphia (ANSP, now "of Drexel University") crews led by Dr. Ted Daeschler. All necessary permits were obtained for those prior fossil collection campaigns, which complied with all relevant regulations [25–27]. For this study, fossil specimens were selected, in part, to achieve a diverse sampling of Devonian "placoderm", stem chondrichthyan, and stem osteichthyan clades (Table A in S1 File). As negative controls, we also acquired and demineralized one rock matrix sample from the fossiliferous horizon at one of the source fossil localities in each of the three geologic formations (Catskill, Fram, Nordstrand Point) studied herein. Fragmented fossil specimens and fossil specimens coated with consolidants (e.g., vinac, butvar, or similar adhesives) were avoided during specimen selection to preclude glue casts as a potential source of any demineralization products.

Seven of the nine fossil bone samples examined in this study derive from the Famennian (~ 361 Ma) Catskill Formation (Table 1 and Table A in S1 File). The Catskill Formation primarily consists of grayish-red sandstone, siltstone, and mudstone deposited in prograding deltas [25]. The rock matrix at each of the three Catskill Formation sites from which the fossils examined herein were collected consists of pale red, very-fine grained sandstone [25]. Indeterminate bothriolepid sample NUFV 1586 was collected from an exposure of the Frasnian (~ 378 Ma) Fram Formation in Nunavut territory, Canada. The Fram Formation primarily consists of well-indurated sandstones and poorly-indurated siltstones deposited in meandering fluvial channels and adjacent floodplains [26]. Specimen NUFV 1586 was collected from site NV0403, which is one of several Late Devonian sites in the Fram Formation in the southwestern peninsula of Ellesmere Island. Fossiliferous rocks at this locality, and at nearby site NV2K17 from which a representative rock matrix sample was examined, consist of pale gray-red siltstone interpreted to represent a crevasse splay deposit [26]. The specimen of *Psammosteus* sp. examined in this study was collected from the Frasnian (~ 375 Ma) Nordstrand Point Formation in Nunavut territory, Canada. The Nordstrand Point Formation consists of green, pale-red, and gray siltstones interbedded with light-gray, very fine to fine sandstones [27]. The entombing rock matrix at site NV2K11, from which the *Psammosteus* sample was collected, consists of gray siltstone interpreted to have been deposited in a fluvial overbank setting [27].

## Methods

Demineralization assays were conducted in a dedicated laminar flow hood within the former Molecular Paleontology Lab at Rowan University, which was dedicated to biomolecular analyses of fossils and in which modern tissue samples were prohibited (this facility has since been repurposed). Incubation in EDTA causes demineralization via chelation of calcium cations from bone, thus leaving behind iron oxides as well as any organic remains not possessing divalent cations [1].

Using a sterilized chisel, small fragments of cortical bone were collected for the demineralization assays. Disposable polypropylene pipettes (1/well) were used for EDTA solution exchanges. Nitrile gloves, a hair net, lab coat, and face mask were worn by all project personnel while conducting the demineralization assays in the Molecular Paleontology Laboratory. EDTA solution was refreshed every 48–72 hours. Most bone samples required ~ 10 weeks to breakdown into loose "debris", which we collectively refer to in this study as demineralization products (in following of standard practice in this field; e.g., [2]). In contrast, two of the three representative rock matrix samples (specifically those from the NV2K11 and Red Hill localities) were found to be highly resilient to disaggregation in EDTA, so after 14 weeks they were gradually dissolved over a period of one week in 0.2 M HCl (as in [3]).

For imaging of each sample, the tip was cut off a disposable polypropylene pipette, which was then used to transfer two to three drops (~ 50 µl) of concentrated demineralization products to standard glass microscope slides. A glass coverslip was then placed over each slide prior to imaging with the Z-stack feature on a Zeiss AxioLab A1 transmitted light microscope with an integrated AxioCam 506 camera. Slides were initially examined under 10X to find larger, potentially-organic remains (i.e., 'vessels'), then 20X and 40X objectives were primarily used to search for and image small potentially-organic remains (i.e., 'osteocytes' and 'fibrous matrix' fragments). In the absence of biochemical verification of their identities, such purported cells and soft tissue microstructures isolated from the fossils will hereafter be referred to using single quotation marks, in following of common practice in similar prior studies [cf., 8,28,29]. In especially productive samples, a simple counting method was used to estimate the approximate number of microstructures present in each slide. This procedure involved tallying the number of each type of microstructure within each field-of-view area under 20X magnification until the entire area of the covered-slipped sample had been searched.

In addition to transmitted light microscopy, demineralization products from select samples were further examined (without gold or carbon coating) using a variable pressure FEI Apreo field-emission gun scanning electron microscope (FEG-SEM) with an Everhart Thornley Detector and an integrated Oxford AZtec energy dispersive x-ray spectroscopy (EDX) probe for elemental analyses. To concentrate and stabilize the isolated demineralization products for imaging by SEM-EDX, they were rinsed three times in Type I (18.2 Ω) water, transferred as above onto ~ 1 cm² silicon chips (as in [8]), and then left to dry for 48 hrs. Structures morphologically consistent with blood vessels required a magnification of 120–1,500X for imaging, whereas those morphologically consistent with osteocytes and fibrous matrix were best viewed around 2,000–6,500X. Select "close up" images were acquired at 10,000–35,000X magnification. SEM images were collected in secondary electron imaging mode at a working distance of ~10 mm, typically using an accelerating voltage of 10 kV; in a few instances, a lower kV setting was used (down to 2 kV). EDX analyses were performed under these operating conditions at an incident angle of 35° to the stage and over reading times ranging from 25–50 s.

## Results

Examination of demineralization products from the fossil samples revealed abundant microstructures morphologically consistent with vertebrate osteocytes, as well as blood vessels and fibrous extracellular matrix (Fig 1). No similar microstructures were recovered from the three rock matrix samples, which instead yielded subrounded to angular grains of quartz silt and sand and opaque aggregates of silty mudrock (e.g., Fig A in S1 File). Microstructures recovered from the fossils frequently exhibited various shades of light tan/yellow to dark brown coloration, as in numerous previous studies [e.g., 1,3–5,8,14,23,30]. All examined taxa, regardless of their phylogenetic affinities, yielded identifiable microstructures

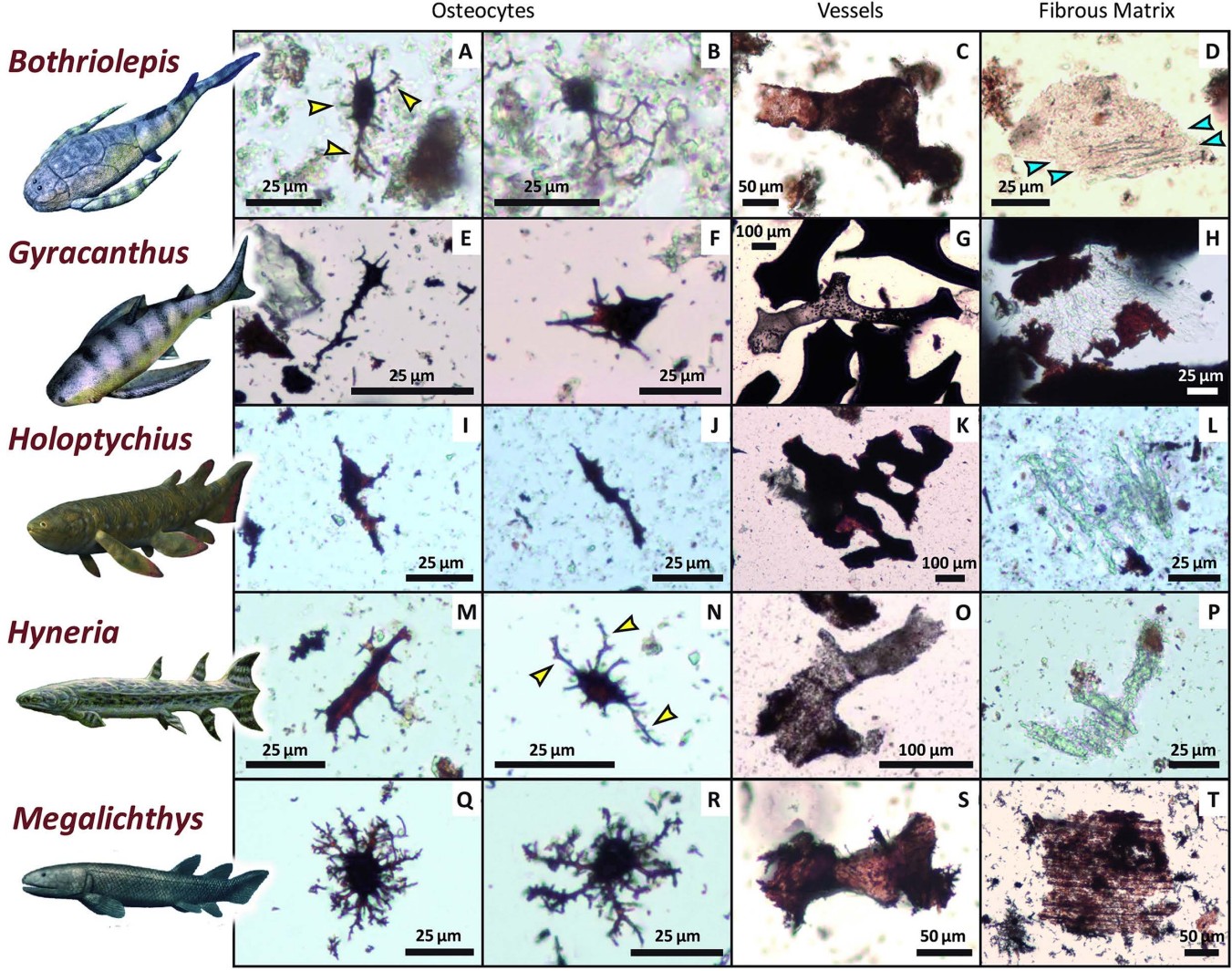

**Fig 1. 'Osteocytes', 'blood vessel' pieces, and fragments of 'extracellular fibrous matrix' isolated from ~375-million-year-old fish bones and scales.** The first two columns display example 'osteocytes', the third shows 'blood vessel' pieces, and the last shows sheets of 'fibrous matrix'. Specimens derive from each taxon by row, as follows: **(A to D)** *Bothriolepis*, **(E to H)** *Gyracanthus*, **(I to L)** *Holoptychius*, **(M to P)** *Hyneria*, **(Q to T)** *Megalichthys*. Select examples of elongate 'filopodia' off of 'osteocyte cell bodies' are noted by yellow arrowheads in **(A)** and **(N)**. Note linear fabric of the 'fibrous matrix' fragments, with an example noted between the blue arrowheads in **(D)**. Scale bars as indicated. Fish reconstruction drawings by Nobu Tamura and ABelov2014 (published under CC-BY-SA 3.0 licenses).

morphologically consistent with vertebrate cells and soft tissues (Table 1 and Table B in S1 File), supporting the hypothesis that taxonomic identity does not exert a meaningful influence on the extent or style of cellular and soft-tissue preservation [5,21].

The most commonly recovered organic microstructures consisted of a prolate spheroid 'core' surrounded by short to long, thin branching strands. These structures are morphologically consistent with vertebrate bone cells (osteocytes), which are comprised of a central cell body with emanating filopodia (e.g., figure 1a in [21]). Most recovered 'osteocytes' were flattened-oblate in shape (*sensu* 30; with a fusiform 'cell body'), though stellate [*sensu* 30] forms (with a spherical 'cell body') were also common, with the central 'cell body' ranging in greatest length from 10–50 µm. 'Filopodia' possessed

up to quaternary ramifications (e.g., Fig 1B and 1Q) and were observed to extend from the 'cell body' by up to 120 µm (especially in specimen NUFV 1586). 'Osteocytes' varied widely in color, but most appeared either semitranslucent orange-red, very dark brown, or transparent and colorless, with all three color variants occasionally found within the same sample (in NUFV 1586 and *Hyneria*, similar to the results of [3]).

'Osteocytes' recovered from *Bothriolepis*, the first sample to breakdown, typically retained long, branching 'filopodia', whereas most 'osteocytes' recovered from *Holoptychius* and *Gyracanthus* lacked apparent 'filopodia'. 'Filopodia' were also occasionally encountered as intricate, broken pieces or as entangled bundles in demineralization products from sample NUFV 1586. This specimen also yielded several stellate and flattened-oblate 'osteocytes' with a dark brown center, which, based on its size and location, appears potentially morphologically consistent with a nucleus (Fig 2B and 2E; [cf., 4,30–32]). Several 'osteocytes' recovered from NUFV 1586 and *Hyneria* exhibited sharp transitions from deep red to colorless within the 'cell body' and along 'filopodia' (e.g., Fig 2A and 2C). *Hyneria* and *Megalichthys* yielded the most abundant 'osteocytes' of the taxa examined (i.e., >200 observed in each slide), most of which retained intricate 'filopodia' (Fig 1M, 1N, 1Q, and 1R). *Holoptychius* yielded the fewest 'osteocytes' other than *Psammosteus*, which yielded none. This latter result was expected, as *Psammosteus* possessed acellular bone [33].

SEM observation revealed 'osteocytes' recovered from the bone samples to exhibit a diverse suite of surface textures, ranging from smooth to rough and granular. Crisscrossing linear grooves, potentially representing imprints of collagen

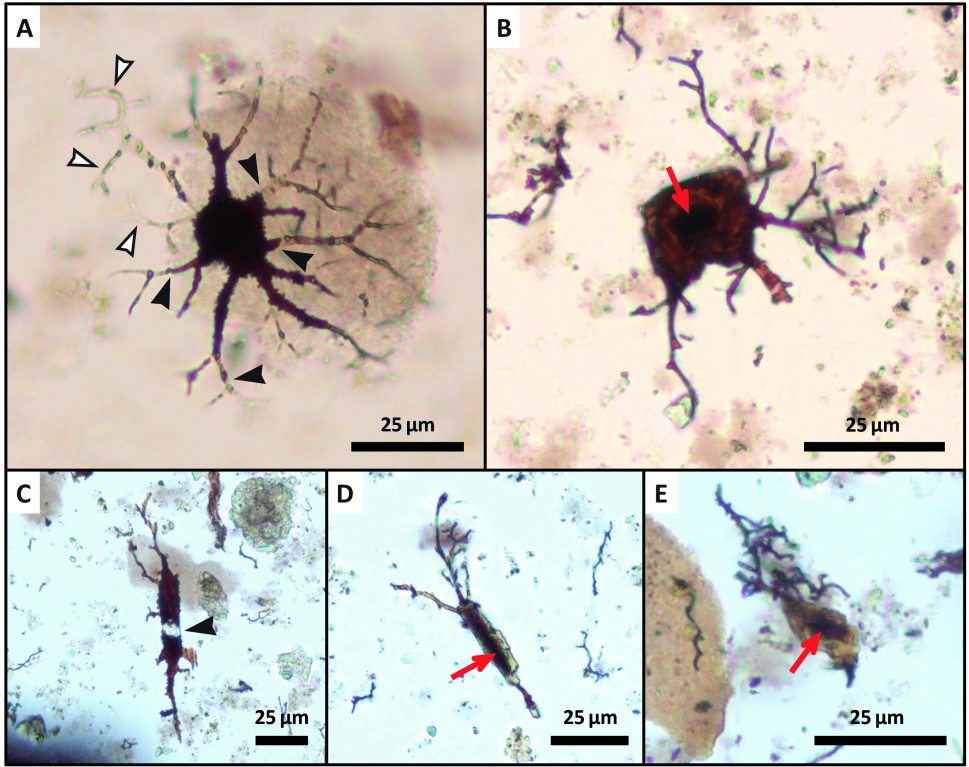

**Fig 2. 'Osteocytes' which display internal heterogeneity, isolated from an anteroventrolateral plate of an indeterminate bothriolepid (NUFV 1586). (A)** An oblate 'osteocyte' partially embedded in a "hazy" fragment of 'extracellular matrix'. Free-floating, semi-translucent 'filopodia' are denoted by white arrowheads, with exemplar transition points between red-orange and semi-translucent colorations denoted by black arrowheads. **(B)** Another oblate 'osteocyte' exhibiting a dark center (red arrow). **(C)** A stellate 'osteocyte' exhibiting a semi-translucent region within its 'cell body', similar to several 'filopodia' of the 'osteocyte' shown in **(A)**. **(D and E)** Two additional stellate 'osteocytes' which possess dark centers (red arrows), as in the 'osteocyte' shown in **(B)**. Scale bars as indicated.

fibrils [5,8,34], were commonly seen on the surface of rough-textured 'osteocytes' (e.g., Fig 3A and 3B). High magnification (~35,000X) of one exceptionally-preserved *Megalichthys* 'osteocyte' revealed the surficial grooves to be recessed into a complex ultrastructure of minute, ~100 nm, blade-like crystallites of iron oxide (see below) oriented at various directions within patches across the 'cell body' (Fig 3B). Alternatively, the 'cell bodies' of a few 'osteocytes' recovered from *Bothriolepis*, *Megalichthys*, and *Hyneria* were coated by larger, ~1–5 µm, plate-shaped crystals of iron oxide (see below).

Cylindrical microstructures morphologically consistent with blood vessel fragments were found in all nine fossil samples (e.g., Fig 1C, 1G, 1K, 1O, and 1S). These fragments varied in length from ~100–1,400 µm with diameters typically ~20–100 µm. Many 'vessels' appeared hollow with thin walls, though 'vessels' apparently infilled with, or casted by, opaque mineral(s) [cf., 8] were also common. Branching (e.g., Fig 1G and 1O) and anastomosing (e.g., Fig 1K) morphologies were both commonly seen, including up to tertiary ramifications. Hollow, semitranslucent 'vessel' fragments ranged in color from light gray to dark red-brown, with most exhibiting a reddish hue. This style of preservation was especially common in *Bothriolepis* and *Megalichthys* (e.g., Fig 1C and 1S). *Holoptychius* and *Gyracanthus* yielded the most abundant 'vessel' fragments, while *Bothriolepis*, NUFV 1586, and *Psammosteus* yielded very few 'vessels' (Table 1). Most 'vessel' fragments recovered from these taxa appeared weakly transparent and dark red. Recovery of 'vessel' fragments but not of 'osteocytes' from the

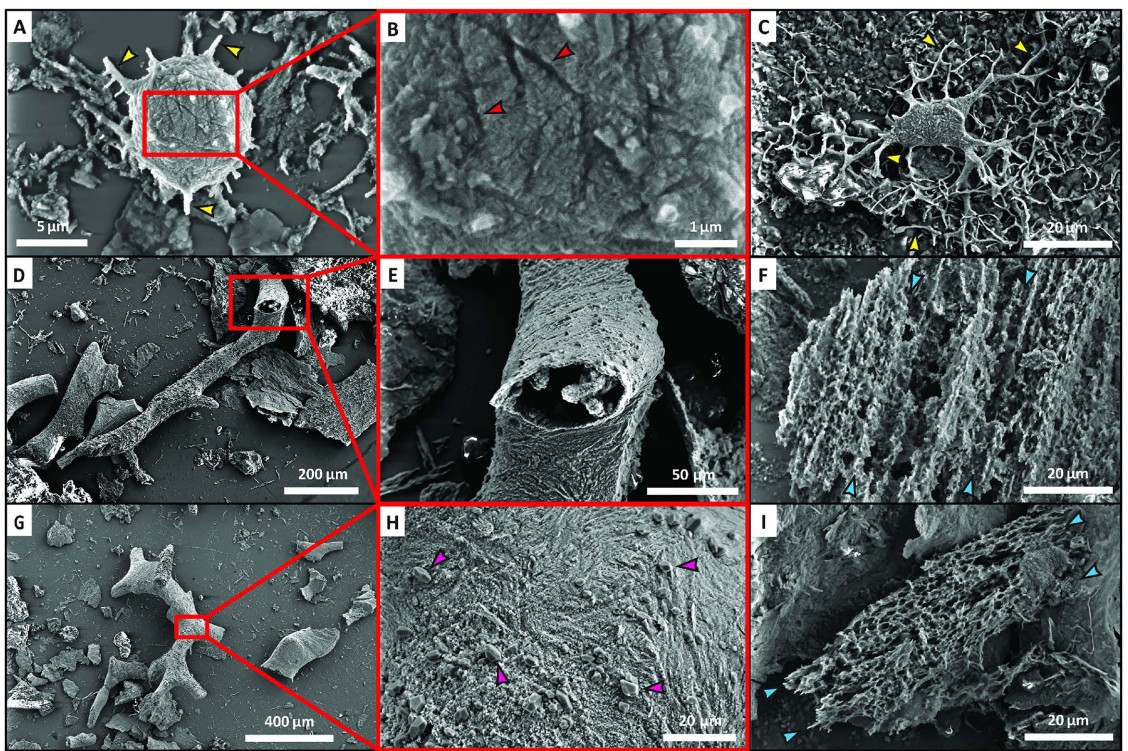

**Fig 3. Scanning electron micrographs of 'osteocytes', 'blood vessel fragments', and fragments of 'fibrous matrix' recovered from Devonian fish scales and bones. (A and C)** Two oblate 'osteocytes' retaining minute 'filopodia' emanating from the 'cell body' (yellow arrowheads), which were recovered, respectively, from a scale of *Megalichthys* and an anteroventral plate of an indeterminate bothriolepid (NUFV 1586). **(B)** Higher magnification of the 'cell body' in **(A)**, revealing crisscrossing grooves (red arrowheads) across its surface. **(D and G)** Two cylindrical 'blood vessel' fragments recovered from a fin spine of *Gyracanthus*. **(E and H)** Higher magnification images of the 'vessels' shown in **(D and G)**, revealing their alternative hollow **(D)** and permineralized **(G)** forms and smooth yet fibrous surface textures. The permineralized vessel cast shown in **(G and H)** also possesses a sporadic coating of quartz crystals ~1–5 µm in size (pink arrowheads). **(F and I)** Two sheets of rough, fibrous, 'extracellular matrix' recovered, respectively, from a scale of *Holoptychius* and an anteroventrolateral plate of an indeterminate bothriolepid (NUFV 1586). The blue arrowheads in **(F and I)** denote the linear fabric of each 'tissue' fragment. Scale bars as indicated.

specimen of *Psammosteus*, despite an equally exhaustive search, is consistent with the vascular yet acellular nature of bone in this heterostracan [33]. 'Vessel' pieces recovered from *Bothriolepis* appeared more brittle and fractured than those from the other fossil specimens. All samples which yielded dark 'vessel' casts, namely those of *Gyracanthus*, *Holoptychius*, and *Megalichthys*, also yielded light gray, semitranslucent 'vessels' with a "speckled" appearance (e.g., Fig 1G).

When viewed via SEM, 'vessel' surface textures typically appeared rough, fibrous, and/or microgranular (e.g., Fig 3D and 3E). A minority of 'vessel' fragments, primarily permineralized casts, exhibited a more jagged texture resulting, in part, from a sparse coating of minute, euhedral crystals (e.g., Fig 3H; see below). At high magnification, both hollow and permineralized 'vessel' fragments commonly exhibited a fabric of curvilinear ridges and grooves across their external surfaces (e.g., Fig 3E).

Although fibrous (proteinaceous) matrix represents the most abundant organic microstructure within cortical bone during life [35,36], 'matrix' fragments were the least common type of potentially-endogenous microstructure found. 'Fibrous matrix' fragments were identified as mesh-like sheets of semitranslucent 'tissue' (e.g., Fig 1D and 1L) typically ~60–160 μm in greatest dimension (though pieces > 300 μm in length were also occasionally observed). 'Matrix' fragments ranged from essentially colorless (in all taxa except *Megalichthys*) to light to dark yellow (in NUFV 1586) or red-brown (primarily in *Megalichthys*; e.g., Fig 1T). The *Psammosteus* sample yielded all three of these color variants. A number of colorless 'fibrous matrix' sheets recovered from *Gyracanthus* were partially surrounded by semitranslucent, red-brown, erratically-shaped "patches" of remaining bone mineral (e.g., Fig 1H). A few pieces of 'matrix' contained embedded 'osteocytes' (primarily in sample NUFV 1586), but this was rarely observed. *Bothriolepis* and *Psammosteus* were found to yield the most 'matrix' pieces, yet even these samples each fell within the Uncommon category (Table 1). The *Hyneria* sample yielded only very small fragments of 'fibrous matrix'.

Under SEM, 'matrix' pieces were found to be of random shapes characterized by a uniform lattice structure composed of a linear fabric of rough, parallel-oriented 'fibers' (e.g., Fig 3F and 3I). High magnification revealed such fragments to possess minute, pore-like, open spaces between long, anastomosing 'fibers'.

Energy dispersive x-ray spectroscopy (EDX) analyses revealed many of the isolated cellular and soft-tissue microstructures to possess elemental profiles potentially suggestive of partial organic (i.e., endogenous) composition. This conclusion is primarily based on carbon (C) content, which was frequently found to be > 5 wt.% and occasionally > 10 wt. % (e.g., Fig 4C and 4I). The highest observed weight percentages of C occurred in several 'vessel' fragments, which presented C > 20 wt.% (e.g., Fig 4F). It is important to note, however, that these weight percentages for C (and nitrogen [N] and oxygen [O]) should be taken as rough estimates due to the difficulty of acquiring accurate measures of the weight percentages of low atomic number elements via EDX (due to their low photon energy, as well as potential peak shifts and interferences with L-, M-, and N-shell x rays of heavier elements [37]) and the potential for minor inflation of C signals stemming from residual organic gas(es) and/or grease within the vacuum chamber of an SEM during analyses [38].

Generally speaking, all three recovered microstructure types ('osteocytes', 'vessels', and 'fibrous matrix') exhibited similar elemental compositions. Both point analyses (Fig 4) and elemental maps (Figs B-D in S1 File) identified most isolated 'cells' and 'soft tissues' to be primarily composed of iron (Fe), O, and C, with varying minor amounts of calcium (Ca), N, aluminum (Al), and sodium (Na). A minority of microstructures, most of which were 'vessels', were also found to exhibit weight percentages for Si exceeding 10 wt. % (e.g., Fig 4D-F); however, these Si readings appear to likely stem from sparse diagenetic coatings of micron-size quartz crystals (as in Fig 3H, and see below). In contrast, as in several previous EDX studies of fossil bone demineralization products [5,8], Fe and O were found to frequently comprise over 50 wt.% of each microstructure, occasionally even over 90% (e.g., several 'osteocytes' recovered from *Megalichthys*). Repetition of our initial 10 kV analyses at higher and lower kV settings (15 kV and 5 kV, respectively) did not drastically alter the weight percentage readouts for Fe. Specifically, these re-analyses found Fe to vary in weight percent by only 1.6−7.5 wt.%, around an average difference of only 2.7 wt.%. Thus, we designate the 10 kV Fe readings to be accurate. Magnesium (Mg), potassium (K), and phosphorus (P) were generally observed to be present in low weight percentages (Fig 4).

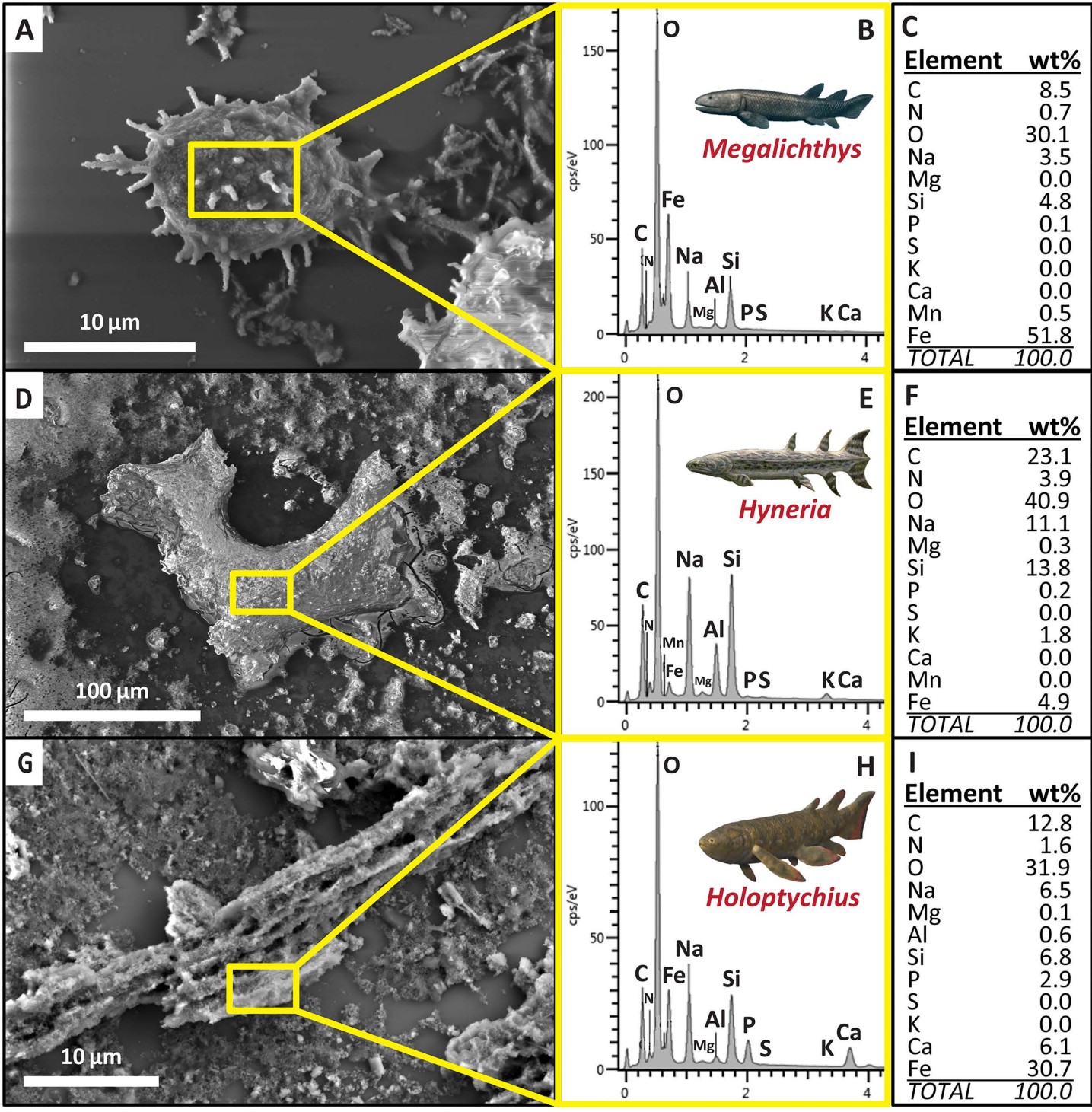

**Fig 4. Scanning electron microscopy-energy dispersive x-ray spectroscopy (SEM-EDX) analyses of exemplar cellular and soft tissue micro-structures recovered from Devonian fish scales and bones. (A to C)** Elemental composition results of EDX on an 'osteocyte' recovered from a scale of *Megalichthys*. **(D to F)** Elemental composition results of EDX on a 'blood vessel' fragment recovered from a dentary of *Hyneria*. **(G to I)** Elemental composition results of EDX on a shard of 'fibrous matrix' recovered from a scale of *Holoptychius*. Note the general abundance of iron (Fe) and oxygen (O) in each sample, as well as noteworthy weight percentages for carbon (C), potentially suggestive of partial organic composition. Scale bars as indicated. Fish reconstruction drawings by Nobu Tamura and ABelov2014 (published under CC-BY-SA 3.0 licenses).

Among the three potentially-endogenous microstructure classes identified, 'osteocytes' were found to possess the most variety in elemental composition. Specifically, the 'cell bodies' of most 'osteocytes' were characterized primarily by high signals for Fe and O (e.g., Fig B in S1 File), indicating the probable presence of goethite [cf., 20,39]. However, a few *Hyneria* 'osteocytes' instead yielded high signals for Si and O (e.g., Fig D in S1 File). 'Osteocytes' of this latter type exhibited a smoother surface texture and lacked elongate 'filopodia', which, in combination with the EDX results, implies that they likely represent quartz casts of cellular lacunae. Where quartz-permineralized casts of osteocyte lacunae have been identified previously, they have been found to possess very short filopodial extensions (e.g., figure 4b in [40]), most likely due to incomplete permineralization of canaliculi and/or breakage of these extremely thin and brittle projections during the preparation of demineralization products for visualization by transmitted light microscopy and/or SEM.

Most fragments of 'blood vessels' and 'fibrous matrix' conform to the common trend of high weight percentages for Fe and O described above, but Al, Na, and K were found to be slightly more abundant in 'vessels' compared to 'fibrous matrix' and 'osteocytes'. Yet, in every case, each of these elements remained < 16 wt.% (e.g., Fig 4F). Some 'vessels' recovered from *Gyracanthus* are coated by a sparse veneer of small (~2–5 μm size), Si-rich crystals of euhedral form consistent with quartz (e.g., Fig 3H). 'Fibrous matrix' fragments frequently exhibited a few more weight percent Ca than 'osteocytes' and 'vessels' (e.g., Fig 4). This trend appears to arise from minor retention of undemineralized bone apatite within the very rough textures of these fibrous 'tissue' fragments (Fig C in S1 File).

## Discussion

### Extending the record

Microstructures that are morphologically consistent with 'osteocytes', intricately-branching 'blood vessels', and mesh-like fragments of 'fibrous matrix' have each previously been recovered from demineralization products of bones from a wide variety of extinct vertebrates, including nonavian dinosaurs, mammals, birds, turtles, fish, and marine reptiles dating as far back as the Permian [21]. Here, we have identified this style of preservation in a diverse suite of Late Devonian fish, none of which had previously been examined by demineralization, namely an acanthodian, heterostracan, antiarch "placoderm", and several basal osteichthyans. These discoveries extend the temporal record of this type of exceptional preservation of cellular and soft-tissue micromorphology [*sensu* 41] back by approximately 100 million years, supporting the inference [22] that this style of cellular/soft-tissue preservation may be independent of geologic age. Our findings also complement those of Ullmann and Schweitzer [21] who found negligible statistical support for preferential preservation of these cellular and soft tissue microstructures across geologic time.

This study also extends the record of this style of cellular and soft-tissue preservation to several bioapatitic tissue types beyond endochondral/perichondral bone, namely osteodentine (in *Gyracanthus*), orthodentine (in *Psammosteus*, *Gyracanthus*, and *Megalichthys*), and aspidin (in *Psammosteus*). Of these three tissue types, only orthodentine had previously been tested for soft-tissue preservation by demineralization, and neither study that previously examined this question [8,42] recovered any microstructures morphologically consistent with an organic source from teeth (comprised primarily of orthodentine, which lacks osteocytes during life). Consequently, our successful recovery of 'osteocytes', 'vessels', and 'fibrous matrix' from diverse bioapatitic tissue types demonstrates that this style of preservation is also independent of bone/dentine microstructure, which in turn implies that a much larger portion of the vertebrate fossil record may contain informative cellular, soft tissue, and (potentially also) biomolecular data than is currently appreciated.

### Inferred preservation pathways

We infer that our successful recovery of intact cellular and soft-tissue microstructures from fossils up to ~378 My in age is likely the product of initial postmortem circumstances which promoted natural fixation and partial mineralization, followed by chemical stability [cf., 2,11,17]. First, the fossils we examined were all collected from sandy, and therefore

initially-permeable, fluvial sediments, which have been posited to favor soft tissue preservation as they can facilitate drainage of autolytic enzymes away from a carcass [14]. Moreover, the pale to deep maroon color of the "red bed" rocks entombing all but one of the fossils we examined (that from the Nordstrand Point Formation) typically signifies deposition under semi-arid conditions [43] in conjunction with, or followed by, early-diagenetic oxidation [44], the latter of which is known to facilitate preservation of soft tissues through free-radical-induced [20,39] and oxidative crosslinking (i.e., gly-coxidation/lipoxidation) of their component biomolecules [42]. Alongside concurrent precipitation of iron oxides and/or hydroxides (e.g., goethite) onto and within cell membranes [i.e., 8,11,39] and tissue surface layers (i.e., the tunica externa of blood vessels), these processes can rapidly confer durable, lasting stability to organic microstructures, helping them to persist over geologic timescales [20]. Based on our observations, we infer that all three of these processes likely acted in concert during early diagenesis of the Devonian bones we examined. This conclusion is consistent with both the sedimentary context of the fossils and our EDX findings that many of the recovered microstructures are enriched in iron and oxygen (especially for 'osteocytes' and 'fibrous matrix'; e.g., Fig 4C and 4I).

It is important to note that occasional permineralization of osteocyte lacunae by quartz in the *Hyneria* specimen is not surprising, as alternative forms of preservation of cellular features mere microns away from one another has been encountered previously [e.g., 11]. Further, that the permineralizing phase in this case was quartz indicates that the *Hyneria* specimen experienced greater diagenetic alteration than the others we examined, and that it was exposed to relatively more acidic and silica-rich pore fluids during early diagenesis [cf., 40]. Additionally, the greater abundance of Na, K, and Al in 'vessels' observed via EDX maps and point analyses (e.g., Fig 4D–4F) indicates that these microstructures may have been preserved through partial aluminosilicification [cf., 11] rather than permeation (or casting) with goethite. Cumulatively, then, our results are not only consistent with prior hypotheses about where this type of exceptional preservation may be encountered, but also support the use of the following taphonomic "search criteria" for future studies on soft tissue preservation within fossil bones: burial in initially sandy/permeable sediments [14], in oxidizing environments [42], and/or in association with abundant iron at the micro-scale [20,39].

## Implications for taphonomy and molecular paleontology

Finally, it is important to note how, based on current knowledge, it is clear that 'blood vessels' and 'osteocytes' can potentially be recovered from a diverse array of bioapatitic tissues in fossil bones of any extinct vertebrate which possessed them, specifically since the origin of vascularized bone in pteraspidomorphs in the Early Ordovician [33] and cellular (osteocytic) bone in osteostracans in the early Silurian [45] respectively. Taphonomically, this implies that the early-diagenetic chemical reactions facilitating this type of exceptional preservation described above (i.e., free radical-induced crosslinking [20], sulfurization [46], aluminosilicification [11], and/or glycoxidation/lipoxidation [42]) can operate within most, if not all, bioapatitic tissues, and that they were likely operating as soon as these biomineralized tissues evolved.

These insights have implications for the value of the vertebrate fossil record, which paleontologists now recognize to extend to the molecular level. Although it is probable that the average severity of diagenetic alteration generally increases with geologic age [47,48], our findings corroborate those of numerous prior reports [e.g., 1–5,8,14,22,30,32] that have repeatedly found 'osteocytes', 'blood vessels', and 'extracellular matrix' to be remarkably resilient against the long-term rigors of diagenesis. Because each of these microstructures are, and always were, composed of a rich suite of proteins and other biomolecules during life, many of which also have high preservation potential in their own right (e.g., collagen I, actin, tubulin [15,49]), much can potentially be learned from biomolecular analyses of such fossil cells and soft-tissue microstructures. In fact, a growing number of studies have revealed how biochemical (and even proteomic) remnants occasionally identified within fossil 'osteocytes', 'vessels', and similar microstructures can shed light on diverse aspects of the physiology, phylogenetic relationships, and evolutionary history of extinct vertebrates. For example, collagen I and actin peptide sequences recovered from fossilized 'extracellular matrix' and 'blood vessels', respectively, have been used to evaluate the phylogenetic relationships of extinct taxa independent of skeletal morphology [13–16]. Similarly, fossil

'osteocytes' have been found to be reservoirs of information not only on an individual's relative age at death [50], but also on bone growth rate [51] and genome size and evolution [e.g., 52,53]. Recognition that 'osteocytes' and 'soft tissues' can be recovered from fossil bones dating back to the middle Paleozoic implies that each of these types of histologic, paleogenomic, and molecular-phylogenetic analyses could potentially yield informative results from exceptionally-preserved fossils of the earliest vertebrates – an intriguing possibility that would have been considered unfathomable just a few short decades ago.

## Supporting information

**S1 File. Supplemental details on demineralization assay results, including four supplemental figures (Figs A-D), two supplemental tables (Tables A and B), and supplementary references.**
(DOCX)

## Acknowledgments

We thank T. Daeschler from the Academy of Natural Sciences of Drexel University for providing the fossil specimen fragments for this study and for granting permissions for destructive analyses, as well as T. Scabarozi, W. Xue, and J. Hettinger from the Rowan University Department of Physics and Astronomy for training and assistance in running the FEG-SEM and EDX analyses. We also thank reviewers E. Cadena and E. Boatman for their constructive comments that strengthened the manuscript.

## Author contributions

**Conceptualization:** Paul V. Ullmann.

**Funding acquisition:** Paul V. Ullmann.

**Investigation:** Christopher L. Rogoff, Paul V. Ullmann.

**Methodology:** Christopher L. Rogoff, Paul V. Ullmann.

**Project administration:** Paul V. Ullmann.

**Visualization:** Christopher L. Rogoff, Paul V. Ullmann.

**Writing – original draft:** Christopher L. Rogoff, Paul V. Ullmann.

**Writing – review & editing:** Christopher L. Rogoff, Paul V. Ullmann.

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
