## [Decision Letter · Decision Letter 0]

22 Aug 2025

Dear Dr. Ullmann,

Thank you for submitting your manuscript to PLOS ONE. After careful consideration, we feel that it has merit but does not fully meet PLOS ONE’s publication criteria as it currently stands. Therefore, we invite you to submit a revised version of the manuscript that addresses the points raised during the review process.

We look forward to receiving your revised manuscript.

Kind regards,

Furqan A. Shah

Academic Editor

PLOS ONE

Journal Requirements:

2. In your manuscript, please provide additional information regarding the specimens used in your study. Ensure that you have reported human remain specimen numbers and complete repository information, including museum name and geographic location.

For more information on PLOS One's requirements for paleontology and archeology research, see https://journals.plos.org/plosone/s/submission-guidelines#loc-paleontology-and-archaeology-research .

3. Please amend your authorship list in your manuscript file to include author Paul Ullmann.

4. Please amend the manuscript submission data (via Edit Submission) to include author Paul V. Ullmann.

Reviewers' comments:

Reviewer's Responses to Questions

**Comments to the Author**

1. Is the manuscript technically sound, and do the data support the conclusions?

Reviewer #1: Yes

Reviewer #2: Partly

2. Has the statistical analysis been performed appropriately and rigorously?

Reviewer #1: N/A

Reviewer #2: N/A

3. Have the authors made all data underlying the findings in their manuscript fully available?

Reviewer #1: Yes

Reviewer #2: Yes

4. Is the manuscript presented in an intelligible fashion and written in standard English?

Reviewer #1: Yes

Reviewer #2: Yes

Reviewer #1: Dear authors. I found your manuscript appealing. See my minor comments and suggestions in order to improve the quality of your manuscript. I hope you will find them useful. See the annotated PDF attached.

sincerely your,

Reviewer #2: I appreciated the opportunity to review this manuscript. Overall, the scientific approach is sound. I do have some specific comments related to how you are reconciling your methods with conclusions, that I expect can be readily addressed based on (perhaps) subtle rewording and/or the inclusion of a small amount of additional data that you already have.

1) You comment on the presence of C in the various tissue remnants, observed by EDS. As you note, the values identified are clearly above background, and I fully accept that there is C in these samples - so this is not a challenge to the conclusion of C being present. However, it may help with your argument to explicitly comment on the fact that EDS typically records several % background C levels due to vacuum greases, etc., evaporated into the system (relevant to lines 300-303). I think acknowledging this and perhaps adding a citation relevant to the system you were using and the relevant environmental settings within the system during your experimental work could help support your case here. (You could also do EDS on a blank sample under the same conditions to test for this.)

2) Lines 318-320 say that both types of elemental analyses (EDS spots, maps) support the presence of Si in the tissue remnants. As a reader of your draft, all I can see as supporting evidence are the three sets of %s from EDS spot analyses in Fig 4 and the maps in Supp Fig B - and these, in combination, do not support that claim. Based on your descriptions of these tissue remnants and my own work with them, I interpret them to largely be relatively low-density, partially organic, and in some cases, hollow. As a result, many of them contain a relatively small amount of matter for the electron beam to interact with, which is important at high voltages, longer dwell times, and high current densities, such as those with EDS spot collection. Hence, noting that you deposited the samples on Si wafers, and considering what the sample materials are, it wouldn't surprise me to observe Si in the EDS spots, in particular. My concern is that the electron beam might have been interacting with the Si substrate and returning a Si signal in some cases that is not true to the tissue remnants. This would be my interpretation when I compare these figure sets, where the map set in Supp Fig B shows the small amounts of both the O and C spatially correlated with the osteocyte structure of interest, but no Si (lower dwell times, lower electron density, etc). Perhaps you have other map sets that more clearly show the Si for samples from the same specimen and this was simply the less favorable map for me, as your reader, to use for a comparison across your EDS spot and map data to analyze whether your data support your conclusion of Si. Alternatively, perhaps I could suggest seeing if you can replicate the EDS spot findings specific to Si contents by depositing the tissue remnants on C sticky tape instead (in this case, the C values wouldn't be reliable, of course!).

3) I'm not convinced about the interpretation of the dark central cores in the osteocyte structures as preserved nuclei. Because you have no way to verify with your current methods, perhaps just be sure your language is appropriate (e.g., "possible preserved nuclei"). In a future study, I wonder if FIB might offer an interesting opportunity to section a few of these structures and look for distinct core/shell attributes. Alternatively, there have been some tremendous advances in beamline optics that could possibly allow for spatial analysis, perhaps by nano-FTIR mapping (e.g., LBNL ALS, ANL APS). But on my end, when I look at the images and consider the fact that the structures are round, I feel that I cannot be sure that the central dark core isn't, simply, a region where the round structure is sufficiently thick to obscure light transmission. For example, your arrow in 2D is pointing to a rather long 'nuclear' structure, which seems less probable as a nucleus to me. Alternatively, perhaps in your existing image set you have a sufficient number of photos that you could skim through to see whether you find larger-dimensioned structures from the same specimens that LACK the dark central cores, or even to see whether there is a general trend toward smaller-dimensioned structures lacking cores and larger-dimensioned structures possessing them - these additional lines of consideration with your existing data could, potentially, further support (or refute) your existing conclusion. As it stands, though, my suggestion would simply be to soften the language because your methods have no way of conclusively determining whether these are preserved nuclei.

4) This comment is more specific to ensuring your methodology/findings/conclusions supporting your question/hypothesis. You phrase your hypothesis/conclusions a few different ways, and as a result, I admit I am not entirely certain that I am clear on your ultimate conclusion in the context of your question/hypothesis. I believe you are probing whether or not geologic age alone can be a limiting factor in preservation - here, preservation is your word. I'm not sure if you truly mean preservation or perhaps persistence/longevity, which I would think makes more sense in light of your study design. That is, I believe your study design is essentially adding another 100 my to the depth of time from which fossilized soft tissue remnants have been recovered - that those additional 100 my of diagenesis in at least some cases, but maybe many many cases, have little implication on whether we can recover fossilized soft tissue remnants. I absolutely accept that your study design supports this conclusion, and I am excited to see the findings. But I'm not sure that's the same as geologic age "bearing no influence on preservation style" (paraphrased from line 52). Your words, as I read them, indicate you are trying to answer the question as to whether or not geologic age is related to how things preserve and/or fundamentally whether geologic time poses a limit - but I'm struggling to see how sediment types and species types can relate directly to geologic age and preservation style. I absolutely think there's a strong connection between sediment types (and other early post-mortem conditions) and species types (tissue types, physiology, etc.) (all of which you discuss), and I recognize that some of the earliest works in this area indicated upper limits on protein/soft tissue persistence, which have been 'violated' time and again by studies like yours and mine. But I can't see that one additional report extending the recovery of soft tissue remnants back by 100 my would also be enough to say geologic age has no bearing - afterall, vertebrates existed even before the age of the specimens which your team sampled. Fundamentally, I want to be clear I'm not challenging your findings or conclusions - just the presentation of them and how they're tied into your initial 'hypothesis' or research question (and maybe simply rephrasing as a question instead of hypothesis would be best). Would it perhaps be better to rephrase as EITHER simply "these findings further extend the records of persistence deeper into geologic time and suggest that if a limit exists, it has not yet been found" OR "these findings suggest other factors than geologic time may be more relevant to whether fossilized soft tissue remnants can be recovered from extinct life forms" (e.g., tissue structure and composition, physiology). Perhaps even more simply, I'm not sure you can say "geologic age is not a limiting factor" based on this report extending back known records by another 100 my. The only thing needed to resolve this comment would be rewording of relevant sections, and a bit more specificity in the question you are trying to answer (or can answer).

5) On line 99 you say "each entombing matrix", but you only analyzed three matrix specimens. To respond to this comment, I am only looking for better clarity in the writing in the manuscript. While I do think analysis of the actual entombing matrix (not simply matrix from the same area near where the specimens were recovered) would be ideal, I'm not clear from your writing whether this is in fact how you would describe the matrix specimens or not. Further, when I read "each", I expect to see a matrix specimen for "each specimen", but the numbers don't seem to line up to me. Could you please add some additional language to the few locations that discuss the matrix specimens and/or remove language that is too specific and not accurate to ensure that the text is as accurate as possible so that your reader most clearly knows how to interpret your findings?

My remaining comments are minor and just included as helpful suggestions.

Line 151 - You mention tallying the remnants using a grid overlay, but you have no statistical analyses or counts, etc. Is this perhaps not worth including at all? When I read it, I expected to see the results from the overlay come up somewhere, and they did not.

Line 437 - You have a missing closing parenthesis

I have always been curious about the fibers that appear WITHIN the vessel structures - which you can see in your image Fig 3E. I routinely saw these in different specimens when I worked with fossil vessels, and based on my knowledge of vessel tissues, I was never entirely confident in identifying them. However, seeing them consistently revealed in additional studies, like yours, makes me think they truly are preserved endogenous fibers - elastin in the media? collagen in the adventitia? Curious!

**Do you want your identity to be public for this peer review?** For information about this choice, including consent withdrawal, please see our Privacy Policy

Reviewer #1: **Yes: ** Edwin-Alberto Cadena

Reviewer #2: No

---

## [Author Response · Author response to Decision Letter 1]

4 Oct 2025

Please see our uploaded "Point by Point" Response to Reviewers file for details about the revisions we made to our manuscript and Supporting Information file.

---

## [Decision Letter · Decision Letter 1]

16 Oct 2025

Age is just a number: Examining the preservation of cells and soft tissues in Bothriolepis and other Devonian fish

PONE-D-25-33219R1

Dear Dr. Ullmann,

We’re pleased to inform you that your manuscript has been judged scientifically suitable for publication and will be formally accepted for publication once it meets all outstanding technical requirements.

Kind regards,

Furqan A. Shah

Academic Editor

PLOS ONE

Additional Editor Comments (optional):

Reviewers' comments:

Reviewer's Responses to Questions

**Comments to the Author**

Reviewer #2: All comments have been addressed

2. Is the manuscript technically sound, and do the data support the conclusions?

Reviewer #2: Yes

3. Has the statistical analysis been performed appropriately and rigorously?

Reviewer #2: N/A

4. Have the authors made all data underlying the findings in their manuscript fully available?

Reviewer #2: Yes

5. Is the manuscript presented in an intelligible fashion and written in standard English?

Reviewer #2: Yes

Reviewer #2: (No Response)

**Do you want your identity to be public for this peer review?** For information about this choice, including consent withdrawal, please see our Privacy Policy

Reviewer #2: **Yes: ** Elizabeth M. Boatman

---

## [Editor Report · Acceptance letter]

PONE-D-25-33219R1

PLOS ONE

Dear Dr. Ullmann,

I'm pleased to inform you that your manuscript has been deemed suitable for publication in PLOS ONE. Congratulations! Your manuscript is now being handed over to our production team.

Kind regards,

on behalf of

Dr. Furqan A. Shah

Academic Editor

PLOS ONE